# Exploring Genetic Determinants: A Comprehensive Analysis of Serpin B Family SNPs and Prognosis in Glioblastoma Multiforme Patients

**DOI:** 10.3390/cancers16061112

**Published:** 2024-03-10

**Authors:** Sohaib M. Al-Khatib, Ayah N. Al-Bzour, Mohammad N. Al-Majali, Laila M. Sa’d, Joud A. Alramadneh, Nour R. Othman, Abdel-Hameed Al-Mistarehi, Safwan Alomari

**Affiliations:** 1Department of Pathology and Laboratory Medicine, Faculty of Medicine, Jordan University of Science and Technology, Irbid 22110, Jordan; 2Faculty of Medicine, Jordan University of Science and Technology, Irbid 22110, Jordan; analbzour20@med.just.edu.jo (A.N.A.-B.); mnalmajali20@med.just.edu.jo (M.N.A.-M.); lmsad20@med.just.edu.jo (L.M.S.); jaalramadneh20@med.just.edu.jo (J.A.A.); nrothman20@med.just.edu.jo (N.R.O.); 3Johns Hopkins University School of Medicine, Baltimore, MD 21205, USA; aalmist1@jh.edu (A.-H.A.-M.); salomar1@jhmi.edu (S.A.)

**Keywords:** glioblastoma multiforme, single nucleotide polymorphism, serpins

## Abstract

**Simple Summary:**

Our study conducted at King Abdullah University Hospital (KAUH) in Jordan offers a comprehensive exploration of the correlation between specific Single Nucleotide Polymorphisms (SNPs) in the Serpin B family and the prognosis of Glioblastoma Multiforme (GBM) patients. We reveal that individuals with the G/T genotype of the rs4940595 (*Serpinb11*) SNP experience worse prognostic outcomes in Jordan compared to those with the G/G-T/T genotype. Additionally, we introduce a Serpin B-related 5-gene risk score and employ bioinformatics analyses with the TCGA-GBM cohort, highlighting the significant association of the Serpin B family with implications for predicting progression-free survival. Pioneering in investigating Serpinb11 SNPs in the Jordanian population, our study establishes a foundation for future research into targeted therapies and precision medicine, closing the gap between genetic variations and clinical outcomes in the context of GBM.

**Abstract:**

Serpins are serine proteinase inhibitors, with several serpins being overexpressed in cancer cells. Thus, we aim to analyze the single-nucleotide polymorphism (SNP) of *Serpinb11* and its association with GBM survival. A cohort of 63 GBM patients recruited from King Abdullah University Hospital in Jordan underwent polymorphism analysis and overall survival (OS) assessments. The Cancer Genome Atlas (GBM) cohort was useful for validation. We constructed a risk score using the principal component analysis for the following Serpin genes: *Serpinb3, Serpinb5, Serpinb6, Serpinb11,* and *Serpinb12*, and patients were grouped into high- vs. low-risk groups based on the median cutoff. Univariable Cox models were used to study the survival outcomes. We identified a significant association between rs4940595 and survival. In the TCGA cohort, *Serpinb3* alterations showed worse OS. Univariable Cox showed worse PFS outcomes with higher SERPINB5 and SERPINB6 expression. A Serpin B 5-gene risk score showed a trend towards worse PFS in the high-risk group. Upregulated DEGs showed GO enrichment in cytokine regulation and production, positive regulation of leukocyte activation, and the MAPK cascade. The high-risk group showed a significantly higher infiltration of M2 macrophages and activated mast cells. Our findings showed a significant role of the Serpin B family in GBM survival in the Jordanian population.

## 1. Introduction

Cancer poses a significant health challenge, ranking as the primary cause of death among individuals aged 40–79 for both males and females. Brain tumors, characterized by elevated mortality and morbidity, are a substantial health concern. More than 15,000 fatalities per year in the United States are attributed to malignant primary brain tumors [1,2]. The yearly incidence of primary malignant brain tumors stands at around 7 per 100,000 individuals, with an age-related increase. The five-year survival rate is approximately 36%. Glioblastomas account for roughly 49% of malignant brain tumors, while diffusely infiltrating lower-grade gliomas constitute 30% [3,4].

Glioblastoma multiforme (GBM), classified as a WHO grade 4 glioma, stands as the most prevalent malignant primary brain tumor and is acknowledged as the most lethal form of malignant brain tumor. The updated classification of CNS tumors designates GBM as the most aggressive adult tumor. In Jordan, there has been a notable 105.9% increase in the incidence rate of GBM between 1990 and 2019, with an age-standardized incidence rate of 4.4 per 100,000 and a prevalence rate of 15.8 per 100,000 [5].

The current guidelines for treating individuals recently diagnosed with GBM involve maximum safe surgical removal, followed by a combination of radiotherapy and concurrent/adjuvant chemotherapy [6]. Complete resection has demonstrated a greater likelihood of better survival and absence of progression compared to partial resection or biopsy [7]. In the event of GBM recurrence, potential treatment choices encompass supportive care, reoperation, re-irradiation, systemic therapies, and combined modality therapy. The significance of reoperation in this context is currently not well defined [8]. Current research indicates that the resectability of a tumor may be influenced by its biological characteristics. The primary impediments to the standard of care include various resistance mechanisms, the immunosuppressive microenvironment, and tumor infiltration [9]. Advancing the standard of care in GBM management by exploring combined therapeutic strategies and delivery methods, encompassing immunotherapy, synthetic molecules, natural compounds, and the inhibition of glioma stem cells, can have the potential to enhance standard therapy in GBM management. [10,11].

A thorough comprehension of the interactions among multiple SNPs within a genomic context is crucial. Exploring the combined effects, referred to as epistasis, and their contribution to the variability in complex traits or diseases can advance our grasp of genetic factors. Moreover, examining the impact of SNPs across diverse populations, particularly among Arab Jordanians, may unveil population-specific associations, addressing potential gaps in personalized medicine and genomic risk assessment [12].

Serine protease inhibitors (Serpins) play a vital role in the regulation of various biological processes, such as inflammation and the immune response. Situated in the 18q21 gene cluster, *Serpinb11* is a polymorphic gene/pseudogene that encodes for a Serpin lacking inhibitory properties [13]. Previous studies suggest that variants of the *Serpinb11* gene affect its inhibitory serpin function and act with a non-inhibitory function [14].

In the context of GBM, there is a vital need for a comprehensive and integrated understanding of the various genes and SNP components involved, as well as their interactions. In this study, we aim to elucidate the connection between the prognosis of GBM patients and the SNPs of the *Serpinb11* gene with the integration of genomic analysis for Serpinb genes, shedding light on the relationship between genetic factors and overall survival in GBM patients of the Jordanian population.

## 2. Materials and Methods

The study was approved by the Institutional Review Board (IRB) of King Abdullah University Hospital (KAUH), Jordan [Institutional Review Board (IRB) code number 6/106/2017, dated 8 June 2017]. All subjects were voluntarily involved and signed a written informed consent. Formal written informed consent from patients was not required with a waiver by the IRB. All clinical investigations were conducted according to the principles expressed in the Declaration of Helsinki consent.

### 2.1. Study Cohort

A cohort of 63 patients diagnosed with GBM and possessing adequate clinical data were enlisted from King Abdullah University Hospital (KAUH), the primary tertiary hospital in northern Jordan, spanning the years 2013 to 2020. GBM diagnoses for all cases were made independently by a pathologist using the 2016 World Health Organization (WHO) Classification of Tumors of the Central Nervous System [15]. Additionally, the study incorporated a total of 226 healthy volunteers who served as controls in this study.

### 2.2. DNA Extraction

The genomic DNA of glioblastoma multiforme (GBM) patients was extracted from formalin-fixed and paraffin-embedded (FFPE) tissue using the commercially available DNeasy Blood and Tissue Kit (Qiagen Ltd., West Sussex, UK), following the manufacturer’s protocols. The quality of the extracted DNA was evaluated through agarose gel electrophoresis and ethidium bromide staining. Additionally, the concentration and purity of the extracted DNA were determined using the NanoDrop 1000^®^ (Thermo Fisher Scientific Inc., Wilmington, NC, USA) spectrophotometer. The identified polymorphisms within the candidate genes were analyzed using the Sequenom^®^ iPLEX assay through sequencing techniques. Table 1 shows SNPs’ positions and primers’ sequences for the *Serpinb11* gene.

### 2.3. Bioinformatics Analysis

To investigate the multi-omics characteristics of the Serpin B family, we carried out a bioinformatics pipeline using the GBM cohort from the Cancer Genome Atlas (TCGA-GBM), including a total of 160 patients with sufficient mRNA gene expression data. Data were accessed and downloaded from the cBioportal database [16,17].

We investigated the gene expression of five members of the Serpin B family, including *Serpinb3*, *Serpinb5*, *Serpinb6*, *Serpinb11*, and *Serpinb12*. A risk score was constructed using principal component analysis (PCA) of the log2-transformed expression of the identified genes by taking the first principal component (PC1), and patients were grouped into high- vs. low-risk groups based on the median cutoff of PC1. Differential expression analysis was performed to identify the differentially expressed genes (DEGs) between high- and low-risk groups using the DESeq2 package [18]. Log2 fold-change (Log2FC) threshold of >1 was set to identify the differentially upregulated genes, and Log2FC < −1 was set to identify the differentially downregulated genes with a false-discovery rate (FDR) corrected *p*-value <0.05 identifying significance. Functional enrichment analysis was performed to identify the enriched gene ontology (GO) terms in the resulting up- and downregulated DEGs using the clusterProfiler package [19]. Gene set variation analysis (GSVA) was carried out to identify the up- and downregulated pathways between high- and low-risk groups using the Hallmarks gene sets from the Molecular Signature Database (MSigDB) [20,21]. We analyzed the immune microenvironment between the high- and low-risk groups using the Cibersort algorithm [22].

### 2.4. Statistical Analysis

In our study, the primary survival outcome examined was overall survival (OS), defined as the duration from surgery to the occurrence of death or the last follow-up for those who remained alive at the time of the final data collection and analysis. The Univariable Cox proportional hazard model was employed to explore the prognostic impact of the identified single nucleotide polymorphisms (SNPs), along with age and sex. Survival rates between different groups were depicted using the Log-rank test and Kaplan–Meier curves. The significance of the association with survival was established at a *p*-value of <0.05. The survival analyses were conducted using the R software package (Version 4.3.1) and involved the utilization of the survminer, survival, and finalfit packages.

For continuous variables, the mean ± standard deviation (SD) was utilized when the data displayed a normal distribution, as confirmed by the Shapiro-Wilk test. In instances where the data did not adhere to normality, the median (Q1, Q3) was employed. Categorical variables were presented using frequencies (percentages %). The correlation between demographic, clinical, and genetic variables with study groups was examined using the Wilcoxon (Mann-Whitney U) test for continuous variables, while the chi-squared (X^2) and Fisher-exact tests were applied for categorical variables when the category count was less than 5. A logistic regression model was fitted to the identified SNPs to analyze the association between genotypes and study groups. The significance level was set at a *p*-value of < 0.05. All analyses were conducted in the R software package (Version 4.3.1) using the glm and gtsummary packages.

## 3. Results

### 3.1. Primary Cohort

Our study encompassed a cohort of 63 glioblastoma multiforme (GBM) patients, comprising 37 males (58.7%) and 26 females (41.3%). The mean age at diagnosis was 50.1 years, with a median overall survival of 2.8 months (range: 0.5–9.9 months), and 30 patients (47.6%) unfortunately succumbed. The average tumor size was 126.7 mm, and nearly half of the patients (49.2%) had tumors located on the right side. A significant proportion (80%, *n* = 48) exhibited liquefactive necrosis, with 35.6% displaying necrosis throughout the entire tumor. Refer to Table 2 for a detailed presentation of the baseline characteristics of the GBM cases included in our study.

The univariable Cox proportional hazard model showed that the codominant model of rs4940595 (*Serpinb11*) showed partial association with survival and a significant difference showing better prognosis in the G/G genotype compared to T/T and G/T genotypes, as shown in Figure 1A. While the G/T genotype of the overdominant model of rs4940595 (*Serpinb11*) showed a significantly worse prognosis in GBM patients compared to the G/G-T/T genotype (HR: 2.75, 95% CI: 1.29–5.88, *p*-value = 0.009), as shown in Figure 1B. Table 3 shows the univariable Cox proportional hazard model analysis of the rs4940595 (*Serpinb11*) SNP in our primary cohort.

### 3.2. Serpin B 5-Gene Risk Score

Using the gene expression data of 160 GBM patients from the TCGA cohort, a Serpin B-related 5-gene risk score was calculated using the first principal component. Table 4 shows the clinical characteristics of the Serpin B 5-gene risk score. IDH mutant patients were associated with the low-risk group (11%), while 60 (98%) of the high-risk group had IDH wild-type (*p*-value = 0.01).

The univariable and multivariable Cox proportional models for OS did not reveal any significant association between the Serpin B 5-gene risk score, as shown in Table 5. While the univariable Cox model for PFS revealed a significant association in the *Serpinb5* gene, showing a significantly worse progression outcome with higher expression of *Serpinb5* (HR: 1.67, 95% CI: 1.15–2.43, *p*-value = 0.007), in addition to higher expression of *Serpinb6* and disease progression (HR: 1.44, 95% CI: 1.06–1.96, *p*-value = 0.021), as shown in Table 5. The 5-gene risk score calculated as PC1 was associated with the partial significance of worse prognosis (HR: 1.27, 95% CI: 1.0–1.61, *p*-value = 0.052), and the low-risk group was partially associated with better disease prognosis (HR: 0.72, 95% CI: 0.51–1.03, *p*-value = 0.073), as shown in Figure 2A.

### 3.3. Bioinformatics Analysis

Differential expression analysis between high- vs. low-risk groups showed 998 significantly upregulated DEGs and 199 significantly downregulated DEGs, as shown in Figure 2B. The full list of DEA results is available in the Appendix A. Gene ontology analysis for the upregulated DEGs showed significant enrichment in the following terms: “positive regulation of cytokine production”, “positive regulation of leukocyte and cell activation”, “positive regulation of MAPK cascade”, “leukocyte chemotaxis”, “regulation of ERK1 and ERK2 cascade” as shown in Figure 2C. While the downregulated DEGs showed significant enrichment in the following GO terms: “DNA-binding transcription activator activity”, “forebrain development”, “negative regulation of neuron differentiation”, “regulation of neuron apoptotic process”, and “GABAergic neuron differentiation” as shown in Figure 2D.

The tumor microenvironment showed a significant difference In monocyte abundance, showing a higher infiltration in the high-risk group (mean: 0.10 vs. 0.07, *p*-value < 0.001), in addition to M2 macrophages with a significantly higher abundance in the high-risk group (mean: 0.52 vs. 0.48, *p*-value = 0.021). Resting mast cells were significantly higher in the low-risk group (mean: 0.06 vs. 0.02, *p*-value < 0.001), while activated mast cells were significantly higher in the high-risk group (mean: 0.04 vs. 0.02, *p*-value = 0.006), as shown in Table 6.

## 4. Discussion

Glioblastoma Multiforme is a rapidly progressing tumor that affects the brain or spinal cord and is the most prevalent form of primary malignant brain tumor in adults. GBM patients have a 5-year survival rate of 7.2% and an average survival duration of 15 months [23,24]. In this study, we examined the rs4940595 (*Serpinb11*) expression in 63 GBM patients. We aimed to investigate the association between rs4940595 (*Serpinb11*) under four different inheritance models and genotypes (G/G, G/T, and T/T) and the survival and prognosis rates of GBM patients in Jordan. This study is the first to explore the rs4940595 (*Serpinb11*) variant and its association with GBM, revealing a significant association in Jordan and worldwide.

In our investigation, it was revealed that the codominant model of rs4940595 (*Serpinb11*) exhibited a partial association with GBM survival with a trend toward a worse prognosis in the G/T genotype. Furthermore, a significant association with a worse prognosis was observed in the overdominant model of patients with the G/T genotype compared to those with the T/T and G/G genotypes. Situated within the 18q21 gene cluster, *Serpinb11* is a polymorphic gene/pseudogene responsible for encoding a non-inhibitory *Serpin* [13]. *Serpinbs* distinguish themselves in various aspects from other *Serpins*. Unlike most *Serpins* that function as extracellular proteins, *Serpinbs* are primarily located within cytoplasmic or nuclear cell compartments. In these compartments, they are believed to play a role in safeguarding against indiscriminate proteolysis [25,26]. Our findings showed a significant association between the G/G genotype of *Serpinb11* and better survival outcomes. The availability of SNP genotyping makes it feasible to test their association with diseases and cancer, potentially revealing a serpin haplotype within clade B that is linked to conditions characterized by substantial changes in the balance between peptidases and inhibitors [27]. Askew et al. indicated that variant residues within the *Serpinb11* framework negatively impacted serpin inhibitory function. Using sodium dodecyl-sulfate polyacrylamide gel electrophoresis (SDS-PAGE) analysis, they showed that reactive site loop (RSL)-cleaved Serpinb11 failed to undergo the stressed-to-relaxed transition typically seen in inhibitory-type serpins [14]. It has been shown previously that non-inhibitory serpins such as *SerpinB5*/Maspin can lead to an increase in the sensitivity of tumor cells to cell death and apoptosis, in addition to preventing migration of tumor cells and cancer metastasis [28].

Particularly, *Serpinb11* has been studied primarily in ovarian cancers; however, this is the first study to examine the role of *Serpinb11* in GBM. A study by Lee et al. investigated the anticancer effects of eupatilin as a potential therapeutic agent directed at *Serpinb11* in ovarian cancer cells, showing an inhibitory effect of eupatilin on *Serpinb11* expression [29]. Eupatilin, a bioactive flavonoid, has gained attention for its anticancer effects [30]. Recent studies suggest that eupatilin exerts its effects by modulating the cell cycle or inhibiting metastatic potential in various cancers, including gastric cancer, endometrial cancer, and glioma cells [30,31,32]. In line with our findings in ovarian cancer, Park et al. showed that higher expression of *Serpinb11* was correlated with a poor prognosis in high-grade serous and clear cell carcinoma of the ovary [33]. These findings suggest the potential utility of *Serpinb11* as a prognostic biomarker [34].

In our bioinformatics analysis, we constructed a 5-gene risk score from the *Serpinb* family, including *Serpinb3, Serpinb5, Serpinb6, Serpinb9, Serpinb11*, and *Serpinb12*. Our results of the *Serpinb* 5-gene risk score revealed a trend toward better progression-free survival in low-risk patients. Furthermore, patients with higher expression of *Serpinb5* and *Serpinb6* were associated with significantly worse outcomes. *Serpinb5*, referred to as Maspin (mammary serine protease inhibitor), was identified as a serine protease inhibitor and recognized as a tumor suppressor, and its loss has been observed in breast and prostate cancers, making it a promising diagnostic marker for monitoring tumor progression [35,36]. Several studies showed that Maspin (*Serpinb5*) functions as a tumor suppressor gene, exerting inhibitory effects on angiogenesis, promoting cellular adhesion, and suppressing the migration of cancer cells [37,38]. In line with our findings, a bioinformatics study by He et al. showed that Serpinb5 expression was upregulated in lung adenocarcinoma and hypomethylated, with associations with poor survival in patients with higher Serpinb5 expression suggesting its role as a possible therapeutic target [39]. Additionally, a study by Ma et al. showed that Serpinb5 mRNA expression was downregulated in glioma with a negative correlation with tumor grade compared to normal brain tissue [37].

Furthermore, *Serpinb6*, previously known as proteinase inhibitor 6 (*PI6*), acts as a universal inhibitor of granule protease. Its expression is widespread, and it plays a role in inhibiting both metastasis and tumor progression [40,41]. A study by Burgener et al. showed that *Serpinb6* inhibits Cathepsin G in neutrophils and monocytes, preventing programmed necrosis [42]. Consequently, Song et al. demonstrated the potential role of *Serpinb6* as a contributor to the regular functioning of CAR-T cells. However, additional research is necessary to validate this concept [43]. Among the other Serpins studied in the literature, *Serpinb3* showed a suppressor of lysosomal-mediated cell death in glioblastoma cancer stem cells. Lauko et al. study illustrates that *Serpinb3* impedes the activity of cathepsin L released from lysosomes, resulting in enhanced resistance to radiation. Targeting this axis could represent a strategy to enhance the effectiveness of radiotherapy not only in glioblastoma but also in other cancer types [44].

Additionally, *Serpinb12*, previously identified as Yukopin, functions as a trypsin inhibitor and exhibits expression in various tissues, including the blood, kidney, liver, heart, and brain. Minimal expression of *Serpinb12* was observed in granular cells, Purkinje cells, and neurons/axons within the cerebellum, as well as in the axons and neuropil of the cerebral cortex [45]. A study by Sun et al. revealed that the gene expression of *Serpinb12* was associated with a protective role in stage I-IIIA lung adenocarcinoma based on recurrence-free survival [46]. Considering protective factors, it has been observed in previous reports that *Serpinb12* exhibits abnormal expression in the lungs. However, there is no additional clarification regarding its specific role [45]. Additionally, animal studies suggest that *Serpinb12* has the potential to serve as a biomarker and may be employed for the early detection of ovarian cancer in women [47].

Our study provides several strengths. First, our sample size is considered sufficient from a clinical perspective due to the low incidence rate of GBM. Second, we utilized a bioinformatics pipeline to explore the multi-omics characteristics of the Serpin B family, involving a GBM cohort from The Cancer Genome Atlas (TCGA-GBM). This adds a layer of complexity to the study, allowing for a comprehensive examination of gene expression, differentially expressed genes, functional enrichment, and pathway analysis.

However, our findings should be interpreted with caution in the context of several limitations. First, our study encountered substantial epidemiological and demographic constraints as it drew cases from a single-center tertiary hospital in North Jordan; thus, the generalizability of the findings is restricted to the Jordanian population. In our study, we have profiled the genotypes of specific SNPs. However, future large-scale multi-center studies incorporating gene sequencing for the whole exome and next-generation sequencing may reveal a significant association with the prevalence and survival rate of GBM cases. Furthermore, we emphasize the importance of conducting additional studies with larger and more generalized samples, representing a broader population. These studies should target multiple factors that were not well studied in this research, potentially providing a better representation of the truth. Integrating additional omics data, such as proteomics and epigenomics, could provide a more comprehensive understanding of the molecular landscape associated with glioblastoma. This could uncover additional therapeutic targets and biomarkers.

## 5. Conclusions

We provided a comprehensive investigation, spanning a substantial cohort from King Abdullah University Hospital (KAUH) in Jordan, showing novel insights into the association between specific SNPs within the Serpin B family and the prognosis of GBM patients. Our findings illustrate worse prognostic outcomes for GBM patients in Jordan with the G/T genotype of the over-dominant model of rs4940595 (*Serpinb11*) SNP compared to those with the G/G-T/T genotype. Furthermore, we developed a Serpin B-related 5-gene risk score, coupled with bioinformatics analyses utilizing the TCGA-GBM cohort, revealing a significant association between the Serpinb family and their implications for progression-free survival prediction. Our study is the first to investigate the role of *Serpinb11* SNPs in GBM within the Jordanian population and bridge the gap between genetic variations and clinical outcomes. We lay the groundwork for future investigations into targeted therapies and precision medicine for GBM.

## Figures and Tables

**Figure 1 cancers-16-01112-f001:**
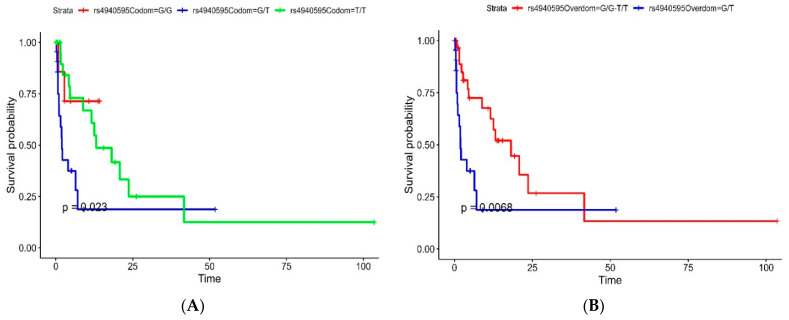
Kaplan–Meier curves of significant survival-associated SNPs. (**A**) The overdominant mode of rs4940595 shows a better prognosis in the G/G-T/T genotype compared to the G/T genotype. (**B**) The codominant mode of rs4940595 shows a worse prognosis in the G/T genotype.

**Figure 2 cancers-16-01112-f002:**
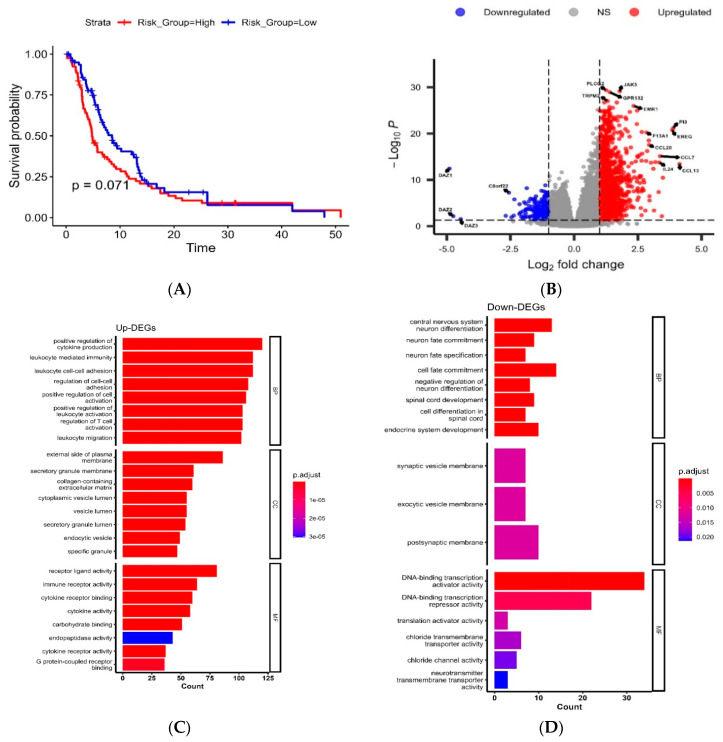
Survival and genomic characteristics in the Serpin B 5-gene risk score in the TCGA-GBM cohort. (**A**) Kaplan–Meier curve for progression-free survival between high- vs. low-risk groups shows a partially significant prognosis in the low-risk group. (**B**) Volcano plot for the differentially upregulated (Log2FC > 1, red) and downregulated (Log2FC < −1, blue) at an FDR-corrected *p*-value < 0.05, grey dots represent insignificant genes (NS). (**C**,**D**) bar plots for the GO analysis for the up- and downregulated DEGs.

**Table 1 cancers-16-01112-t001:** The SNPs, SNPs positions, and primers sequences *Serpinb11*.

SNP-ID	Gene	Chr ^	bp *	Primer Forward	Primer Reverse
rs4940595	*Serpinb11*	18	63,712,604	ACGTTGGATGCTGGAAGAATTCATTCCGAG	ACGTTGGATGTACAGTTAGAGTCTGGCTGG

* bp: base pair (Genomic Position). ^ Chr: Chromosome.

**Table 2 cancers-16-01112-t002:** Baseline characteristics of GBM cases and controls included in our study.

Variable	GBM (n = 63)
Age at diagnosis, Mean (SD)	50.1 (18.4)
Sex, n (%)	
Females	26 (41.3%)
Males	37.0 (58.7%)
Survival Status, n (%)	
Alive	33 (52.4%)
Dead	30 (47.6%)
Overall survival (months), Median (Q1, Q3)	2.8 (0.5, 9.9)
Serum LDH (U/L), Mean (SD)	34.0 (179.0)
Total protein (g/L), Mean (SD)	47.3 (33.1)
Monocytes (×10^9^/L), Mean (SD)	3.7 (4.0)
Lymphocytes (×10^9^/L), Mean (SD)	9.2 (10.0)
Platelets (×10^3^/μL), Mean (SD)	281.9 (96.4)
Tumor size (mm), Mean (SD)	126.7 (96.9)
Tumor laterality, n (%)	
Right	31 (49.2%)
Left	29 (46.0%)
Bilateral	3 (4.8%)
Necrosis, n (%)	
Coagulative	7 (11.7%)
Geographic	1 (1.7%)
Liquefactive	48 (80.0%)
None	4 (6.7%)
Degree of necrosis, n (%)	
Foci of palisading necrosis	34 (57.6%)
Whole tumor	21 (35.6%)
None	4 (6.8%)
Radiotherapy, n (%)	9 (14.3%)
Chemotherapy, n (%)	6 (19.4%)

**Table 3 cancers-16-01112-t003:** Univariable Cox proportional hazard model for overall survival of four modes of inheritance of rs4940595 (SERPINB11) SNP in GBM primary cohort.

SNP ID	Model	Genotype	HR (95% CI, *p*-Value)
rs4940595	Codominant	G/G	-
		G/T	3.87 (0.87–17.26, *p* = 0.076)
		T/T	1.51 (0.34–6.79, *p* = 0.592)
	Overdominant	G/G-T/T	-
		G/T	2.75 (1.29–5.88, *p* = 0.009)
	Dominant	G/G	-
		G/T-T/T	2.25 (0.53–9.56, *p* = 0.271)
	Recessive	G/G-G/T	-
		T/T	0.53 (0.25–1.14, *p* = 0.106)

**Table 4 cancers-16-01112-t004:** Clinical and genomic characteristics between high- and low-risk groups in GBM-TCGA cohort.

Characteristic	High, *n* = 80 ^1^	Low, *n* = 80 ^1^	*p*-Value ^2^
Sex			0.4
Female	16 (36%)	28 (44%)	
Male	28 (64%)	35 (56%)	
Sample Type			>0.9
Primary	76 (95%)	77 (96%)	
Recurrence	4 (5.0%)	3 (3.8%)	
Subtype			0.9
IDHmut	4 (6.3%)	3 (4.7%)	
IDHwt	57 (90%)	59 (92%)	
Fraction Genome Altered	0.20 (0.13)	0.23 (0.14)	0.13
MSIsensor Score	0.31 (1.02)	0.28 (0.32)	<0.001
Mutation Count	57 (64)	216 (1367)	0.2
OS Time (Months)	14 (12)	14 (13)	0.8
OS Status	68 (86%)	59 (74%)	0.053
PFS Time (Months)	8 (10)	9 (8)	0.082
PFS Status	69 (87%)	57 (71%)	0.012
TMB (nonsynonymous)	1.87 (2.14)	7.15 (45.51)	0.2

^1^ Mean (SD); n (%) ^2^ Wilcoxon rank sum test; Pearson’s Chi-squared test; Fisher’s exact test; Abbreviations: IDHmut, IDH-mutant, IDHwt, IDH-wild type, OS, overall survival, PFS, progression-free survival, TMB, tumor mutational burden.

**Table 5 cancers-16-01112-t005:** Cox proportional hazard model for overall survival (OS) and progression-free survival (PFS) and the 5-gene risk score.

Factor	OS Univariable	OS Multivariable
	HR (95% CI, *p*-value)	HR (95% CI, *p*-value)
*SERPINB11*	0.92 (0.18–4.70, *p* = 0.920)	0.75 (0.13–4.19, *p* = 0.741)
*SERPINB12*	0.86 (0.41–1.81, *p* = 0.690)	0.93 (0.43–2.01, *p* = 0.852)
*SERPINB3*	1.13 (0.60–2.10, *p* = 0.705)	1.10 (0.57–2.11, *p* = 0.776)
*SERPINB5*	1.05 (0.69–1.59, *p* = 0.817)	1.02 (0.67–1.56, *p* = 0.925)
*SERPINB6*	1.23 (0.91–1.67, *p* = 0.172)	1.22 (0.89–1.66, *p* = 0.212)
*SERPINB9*	1.07 (0.85–1.35, *p* = 0.571)	1.04 (0.67–1.62, *p* = 0.854)
Risk Score	1.11 (0.88–1.40, *p* = 0.384)	NA (NA-NA, *p* = NA)
Risk Group		
High	Reference	Reference
Low	0.91 (0.64–1.30, *p* = 0.607)	0.98 (0.51–1.89, *p* = 0.951)
Factor	PFS Univariable	PFS Multivariable
	HR (95% CI, *p*-value)	HR (95% CI, *p*-value)
*SERPINB11*	1.61 (0.40–6.48, *p* = 0.505)	1.30 (0.29–5.79, *p* = 0.728)
*SERPINB12*	0.48 (0.16–1.46, *p* = 0.196)	0.49 (0.16–1.57, *p* = 0.232)
*SERPINB3*	1.24 (0.70–2.17, *p* = 0.461)	1.03 (0.56–1.90, *p* = 0.925)
*SERPINB5*	1.67 (1.15–2.43, *p* = 0.007)	1.62 (1.12–2.35, *p* = 0.010)
*SERPINB6*	1.44 (1.06–1.96, *p* = 0.021)	1.30 (0.94–1.79, *p* = 0.107)
*SERPINB9*	1.19 (0.94–1.52, *p* = 0.149)	0.94 (0.61–1.46, *p* = 0.789)
Risk Score	1.27 (1.00–1.61, *p* = 0.052)	NA (NA-NA, *p* = NA)
Risk Group		
High	Reference	Reference
Low	0.72 (0.51–1.03, *p* = 0.073)	0.72 (0.38–1.37, *p* = 0.311)

Abbreviations: OS, overall survival, PFS, progression-free survival, HR, hazard ratio, CI, confidence interval.

**Table 6 cancers-16-01112-t006:** Tumor microenvironment using the Cibersort algorithm in the TCGA-GBM cohort between high- and low-risk groups.

Cells	High, *n* = 80	Low, *n* = 80	*p*-Value
B cells naive	0.006 (0.012)	0.004 (0.007)	0.9
B cells memory	0.012 (0.018)	0.013 (0.017)	0.8
Plasma cells	0.001 (0.003)	0.002 (0.007)	0.6
T cells CD8	0.04 (0.03)	0.05 (0.04)	0.1
T cells CD4 naive	0.0000 (0.0002)	0.0020 (0.0110)	0.2
T cells CD4 memory resting	0.08 (0.05)	0.08 (0.06)	>0.9
T cells CD4 memory activated	0.0020 (0.0086)	0.0001 (0.0007)	0.061
T cells follicular helper	0.023 (0.019)	0.034 (0.035)	0.11
T cells regulatory Tregs	0.009 (0.012)	0.008 (0.011)	0.4
T cells gamma delta	0.002 (0.009)	0.005 (0.015)	0.2
NK cells resting	0.04 (0.04)	0.04 (0.05)	0.5
NK cells activated	0.017 (0.021)	0.021 (0.025)	0.3
Monocytes	0.10 (0.06)	0.07 (0.06)	<0.001
Macrophages M0	0.03 (0.07)	0.06 (0.11)	0.15
Macrophages M1	0.015 (0.019)	0.011 (0.016)	0.043
Macrophages M2	0.52 (0.11)	0.48 (0.12)	0.021
Dendritic cells resting	0.0010 (0.0036)	0.0001 (0.0006)	0.082
Dendritic cells activated	0.0013 (0.0024)	0.0018 (0.0042)	>0.9
Mast cells resting	0.02 (0.04)	0.06 (0.07)	<0.001
Mast cells activated	0.04 (0.06)	0.02 (0.04)	0.006
Eosinophils	0.003 (0.010)	0.004 (0.012)	0.5
Neutrophils	0.028 (0.020)	0.023 (0.019)	0.2

## Data Availability

Primary data involving the GBM cohort from KAUH associated with this study is available upon request from the authors. The validation cohort of TCGA-GBM is available publicly through the Genomic Data Commons Data Portal (GDC (cancer.gov (accessed on 1 January 2024))).

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
