# Peer review of "Exploring Genetic Determinants: A Comprehensive Analysis of Serpin B Family SNPs and Prognosis in Glioblastoma Multiforme Patients"

_cancers, 2024, doi:10.3390/cancers16061112_

Round 1

Reviewer 1 Report

Comments and Suggestions for Authors

1.     In the introduction section, it is better to explain the reasons to select Serpinb11 gene for SNP analysis. It seems that the selection is the results of bioinformatic analysis. If so, the introduction should be rephrased.

2.     In the bioinformatic analysis section, the set of low- and high- risk groups was not clear. Usually, it should be based on the OS or other end events. Why did the authors select one of the PCs?

3.     The discussion section should address to the major findings.

Author Response

Response to Reviewer 1 Comments

We appreciate the respected reviewer comments. The following are our point-by- point responses.

Reviewer 1:

  1. In the introduction section, it is better to explain the reasons to select Serpinb11 gene for SNP analysis. It seems that the selection is the results of bioinformatic analysis. If so, the introduction should be rephrased.
  • Response 1: Thank you for your valuable feedback. We have revised the introduction section to explain the reason we selected Serpinb11 SNP (l.74-87) and (l.94-98).
  1. In the bioinformatic analysis section, the set of low- and high- risk groups was not clear. Usually, it should be based on the OS or other end events. Why did the authors select one of the PCs?
  • Response 2: We appreciate your comment. We did not go for a cutoff based on overall survival or outcomes to avoid potential bias of seeking significant results, as when using OS to determine cutoff point, we make sure that patients are classified into two distinct groups based on their significant differences in survival (outcome-based cutoff). However, our aim was to obtain an objective measure and then investigate its association with survival outcomes, thus principal component analysis (PCA) was used for risk modeling, to transforms the expression of the Serpin genes that were considered as risk factor variables into a new set of composite variables (PCs). These new variables (PCs) are uncorrelated and collectively capture the entire variance present in the original data. Then based on the median cutoff for the first PC which captures the highest variance, we compared between the high- and low-risk groups for survival outcomes.
  1. The discussion section should address to the major findings.
  • Response 3: We have revised the discussion section to highlight the major findings in our study (l.272-284).

Reviewer 2 Report

Comments and Suggestions for Authors

Al-Khatib and colleagues presented a research article aimed at evaluating the impact of a gene polymorphism detected in Serpinb11 gene in the survival of GBM. Overall, the authors try to establish a risk score based on SerpinB family expression and polymorphism affecting Serpinb11. The experimental design appears to be simple and the results obtained need further clarification before considering the manuscript suitable for publication. Please see and address the major comments reported below:

1) The abstract section should be shortened;

2) In the Introduction section, please briefly introduce the main problems related to the management of this tumor, including the chemotherapy used and the lack of effective treatments as well as difficulties in surgical treatments and other relevant clinical features. For this purpose, please see:

- https://doi.org/10.1016/j.cell.2023.02.038

- https://doi.org/10.3390/ijms23137207

- https://doi.org/10.3390/cells8080863.

- https://doi.org/10.3390/jpm11040258

- https://doi.org/10.1002/cnr2.1216

3) In the methods section, you should clarify who are the “healthy controls”. Are they patients hospitalized for other diseases or healthy volounteers? Please, clarify;

4) In the following sentence and in other part of the methods section do not use the future verb tens: “The genomic DNA of glioblastoma multiforme (GBM) patients will be extracted from formalin-fixed and paraffin-embedded (FFPE) tissue using the commercially available...”;                                                                                                      

5) Did you evaluate other Serpin B family members or only the five members reported in the methods section?

6) Please check the error in the following sentence of the Results section: “The average tumor size was 126.7 cm, and nearly half of the patients (49.2%) had tumors located on the right side.”. do you mean 126.7 mm? Please, clarify;

7) Many units of measurement are not specified in Table 2 (e.g. Serum LDH, Total protein, monocytes, etc.);

8) In the methods section, you should better clarify how many polymorphisms were investigated. The analysis of a single polymorphisms could be meaningless. In addition, what is the frequency of this polymorphisms in the general population?;

9) The Serpinb11 variant investigated is associated with a loss or gain of function? It is not clear what is the importance of this polymorphism in the tumor promoting processes of GBM.

Comments on the Quality of English Language

Minor English editing performed by the authors themselves are needed

Author Response

Response to Reviewer 2 Comments

We appreciate the respected reviewer comments. The following are our point-by- point responses.

Reviewer 2:

Al-Khatib and colleagues presented a research article aimed at evaluating the impact of a gene polymorphism detected in Serpinb11 gene in the survival of GBM. Overall, the authors try to establish a risk score based on SerpinB family expression and polymorphism affecting Serpinb11. The experimental design appears to be simple and the results obtained need further clarification before considering the manuscript suitable for publication. Please see and address the major comments reported below:

  1.  The abstract section should be shortened;
  • Response 1: We have revised the abstract to make it shorter (l.42-56).

  1. In the Introduction section, please briefly introduce the main problems related to the management of this tumor, including the chemotherapy used and the lack of effective treatments as well as difficulties in surgical treatments and other relevant clinical features. For this purpose, please see:

- https://doi.org/10.1016/j.cell.2023.02.038

- https://doi.org/10.3390/ijms23137207

- https://doi.org/10.3390/cells8080863.

- https://doi.org/10.3390/jpm11040258

- https://doi.org/10.1002/cnr2.1216

  • Response 2: We have revised the introduction section based on your suggestions (l.74-87).

  1. In the methods section, you should clarify who are the “healthy controls”. Are they patients hospitalized for other diseases or healthy volunteers? Please, clarify;

  • Response 3: Healthy controls were healthy volunteers; we have clarified it in the revised methods section (l.118).
  1. In the following sentence and in other part of the methods section do not use the future verb tens: “The genomic DNA of glioblastoma multiforme (GBM) patients will be extracted from formalin-fixed and paraffin-embedded (FFPE) tissue using the commercially available...”;                                                                                                    
  • Response 4: We have revised the methods section to be presented in the past tense instead of future verb tense (l.120, 123, 125, and 126).

  1. Did you evaluate other Serpin B family members or only the five members reported in the methods section?
  • Response 5: Only the five members reported in the methods section were evaluated based on previous literature and available data.

  1. Please check the error in the following sentence of the Results section: “The average tumor size was 126.7 cm, and nearly half of the patients (49.2%) had tumors located on the right side.”. do you mean 126.7 mm? Please, clarify;

  • Response 6: Thank you for pointing out. We have revised the unit to 126.8 mm (l.178).

  1. Many units of measurement are not specified in Table 2 (e.g. Serum LDH, Total protein, monocytes, etc.);
  • Response 7: We have added all the missing units in Table 2.

  1. In the methods section, you should better clarify how many polymorphisms were investigated. The analysis of a single polymorphisms could be meaningless. In addition, what is the frequency of this polymorphisms in the general population?

  • Response 8: Thank you for your valuable feedback. We investigated several genes polymorphisms, however, we meant to focus on the Serpinb11 gene, and integrate the polymorphisms and its association with survival in our cohort, and then conduct additional analyses through a bioinformatics pipeline to investigate the association of different genomic biomarkers of the Serpinb family in GBM. Thus our study was not only based on Serpinb11 SNP, but also integrated genomic analysis for other Serpinb family members using the TCGA GBM cohort.

  1. The Serpinb11 variant investigated is associated with a loss or gain of function? It is not clear what is the importance of this polymorphism in the tumor promoting processes of GBM.

  • Response 9: Thank you for pointing out. The Serpinb11 polymorphism G>T, is a coding-sequence variant with a stop-codon, it has been shown that Serpinb11 variants result in a loss of function, as indicated by impaired serpin inhibitory activity. This finding is crucial as a loss of function in the Serpinb11 variant may disrupt the balance of peptidase-inhibitor interactions, potentially contributing to the dysregulation observed in GBM tumor promotion.

Round 2

Reviewer 2 Report

Comments and Suggestions for Authors

Dear Authors,

almost all my previous comments were properly addressed.

As a further comment, please substitute ref 6 (this is not a citation but a webpage) with the following ref: https://doi.org/10.1016/j.cell.2023.02.038.

In addition, increase the font size of the gene names in Figure 2B.

Author Response

Response to Reviewer 2 (round 2) Comments

We appreciate the respected reviewer comments. The following are our point-by- point responses.

Reviewer 2:

  1. please substitute ref 6 (this is not a citation but a webpage) with the following ref: https://doi.org/10.1016/j.cell.2023.02.038.

Response 1: We appreciate your comment. Reference been substituted (l. 387)

  1. In addition, increase the font size of the gene names in Figure 2B.

Response 2: We appreciate your comment. Font size of gene names been increased.